# Longitudinal associations between hearing aid usage and cognition in community-dwelling Japanese older adults with moderate hearing loss

**Saiko Sugiura**[1,2]*, **Yukiko Nishita**[3], **Yasue Uchida**[2,4], **Mariko Shimono**[2], **Hirokazu Suzuki**[2], **Masaaki Teranishi**[2,5], **Tsutomu Nakashima**[2,6], **Chikako Tange**[3], **Rei Otsuka**[3], **Fujiko Ando**[3,7], **Hiroshi Shimokata**[3,8]

1 Toyota Josui Mental Clinic, Toyota, Aichi, Japan, 2 Department of Otorhinolaryngology, National Center for Geriatrics and Gerontology, Obu, Aichi, Japan, 3 Department of Epidemiology of Aging, Center for Gerontology and Social Science, National Center for Geriatrics and Gerontology, Obu, Aichi, Japan, 4 Department of Otorhinolaryngology, Aichi Medical University, Nagakute, Aichi, Japan, 5 Department of Otorhinolaryngology, Nagoya University School of Medicine, Nagoya, Aichi, Japan, 6 Ichinomiya Medical Treatment and Habilitation Center, Ichinomiya, Aichi, Japan, 7 Department of Health and Medical Sciences, Aichi Shukutoku University, Nagakute, Aichi, Japan, 8 Graduate School of Nutritional Sciences, Nagoya University of Arts and Sciences, Nisshin, Aichi, Japan

* saikos@ncgg.go.jp

**Data Availability Statement:** Data cannot be shared publicly because of NILS-LSA's rules. Data are available from the National Center for Geriatrics

## Abstract

We investigated the associations between hearing aids (HA) and the maintenance of cognitive function among community-dwelling older adults with moderate hearing loss. A total of 407 participants aged 60 years or older with moderate hearing loss were recruited from the National Institute for Longevity Sciences, Longitudinal Study for Aging (NILS-LSA). Moderate hearing loss was defined as a pure-tone average of 40–69 dB at 500, 1000, 2000, and 4000 Hz of the better ear, according to the definition proposed by the Japan Audiological Society. Cognitive function was evaluated using the four subtests of the Japanese version of the Wechsler Adult Intelligence Scale-Revised Short Forms (WAIS-R-SF): Information, Similarities, Picture completion, and Digit Symbol Substitution (DSST). A longitudinal analysis of 1192 observations with a mean follow-up period of 4.5 ± 3.9 years was performed. The HA use rate at any time during the follow-up period was 31.4%, and HA users were significantly younger (t-test, p = 0.001), had worse hearing (p < .0001) and higher education (p = 0.001), participated more frequently in the survey (p < .0001), and were less depressed ($\chi^2$ test, p = 0.003) than the older adults not using HA. General linear mixed models consisted of the fixed effects of HA use, follow-up time, and an HA use × time interaction term adjusted for age and pure-tone average thresholds at baseline, sex, education, and other possible confounders. HA use showed significant main effects on the scores for Picture completion and DSST after adjustment; scores were better in the HA use group than in the no HA use group. The HA use × time interaction was significant for the Information score (p = 0.040). The model-predicted 12-year slope with centralizing age indicated that the no HA use group showed greater decline over time on Information scores than did HA use group. The slopes did not differ between HA users and non-users for the Similarities, Picture

and Gerontology for researchers who meet the criteria for access to confidential data. HP:https://www.ncgg.go.jp/ri/lab/cgss/department/ep/index.html Address:Department of epidemiology of aging, Center for Gerontology and Social Science, National Center for Geriatrics and Gerontology, 7-430 Morioka, Obu, Aichi 474-8511, Japan TEL:81-562-46-2311 FAX:81 -562-44-8518 E-mail: otsuka@ncgg.go.jp.

**Funding:** This study was supported research grants from the Research Funding of Longevity Sciences from the National Center for Geriatrics and Gerontology (19-10 for RO, 21-18 for RO). https://www.ncgg.go.jp/ncgg-kenkyu/index.html This study was partially supported by research funds from the Japan Agency for Medical Research and Development (19de0107004h0001) for HS. https://research-er.jp/projects/view/1071365. The funders had no role in study design, data collection and analysis, decision to publish, or preparation of the manuscript.

**Competing interests:** The authors have declared that no competing interests exist.

completion and DSST. In conclusion, HA use may have a protective effect on the decline in general knowledge in older adults with moderate hearing loss.

## Introduction

The proportion of older adults in the global population has been increasing with an increase in the average life span, and Japan is one of the countries with the highest aging populations. The global population of older adults is expected to reach 1.4 billion by 2030 and 2.1 billion by 2050 [1]. Thus, it is crucial to maintain healthy aging and decrease the burden of aging. Hearing and cognitive impairments are the most common chronic conditions in older adults. Approximately, one-third of older adults may show hearing impairments [2], while there are over 50 million older adults with dementia worldwide [3].

Many studies have shown a link between hearing impairment and cognitive problems, and some meta-analyses have also shown that hearing impairment is one of the most effective risk factors for dementia or cognitive decline [4, 5]. We investigated whether baseline hearing status was associated with the degree of cognitive change assessed with four neuropsychological subtests during a 12-year follow-up in a Japanese older population, and reported that the rate of change in cognitive performance over time differed significantly depending on the presence or absence of hearing impairment [6].

Hearing aids (HA) are the primary management option for hearing loss, and some studies have reported that the use of HA may prevent cognitive decline in adults with hearing loss [7–9]. However, HA use was found to show no significant preventive effects on cognitive decline in a meta-analysis [10]. In a recent study, Glick et al. suggested that clinical intervention with well-fit HA may promote cortical organization and functioning and provide cognitive benefits beyond the known benefits of HA use on communication [11]. They investigated changes in the high-density electroencephalogram in the group with mild to moderate hearing loss, and showed that use of well-fit amplification reversed cross-modal recruitment of the auditory cortex for visual processing over the subsequent 6 months.

However, many older adults with hearing loss may not use HA. While some patients with hearing loss can use HA without problems, other may choose to not use them or may stop using them because of unrealistic expectations, unsatisfactory sound quality, poor hearing aid handling skills, discomfort, problems with support, psychosocial problems, and other reasons [12]. The HA-ownership rate varies by nation; for example, the rate of HA owners among hearing impaired people in Denmark was up to 50% [13], while the rate in Japan was under 15% [14]. This finding implies that the usage rate may be even lower. The WHO recommended HA for moderate (41–60 dB) and severe (61–80 dB) hearing loss; however, the HA-ownership rate among community-dwelling adults in Japan with over 40 dB hearing loss was 39.0% [15]. There are several reasons for the low HA usage rate, which include the high cost of HA, embarrassment related to the use of HA or the hearing impairment itself, lack of proper information or guidance for wearing HA, and low satisfaction with HA [16]. The lack of subsidies may be one of the reasons for the low ownership of HA in Japan. In Japan, the public HA subsidy system is generally used only for people with severe hearing loss of 70 dB or more.

A thorough understanding of the actual effectiveness of hearing interventions is essential, even in cases with moderate hearing loss. We had previously reported that regular HA use showed a protective effect against cognitive impairment assessed using the Mini-Mental State Examination in those with moderate hearing loss [17]. Thus, we aimed to clarify whether there

were longitudinal differences in cognitive function between the HA use group and the no HA use group in community-dwelling older adults with moderate hearing loss.

## Materials and methods

### Participants

The participants were enrolled from the National Institute for Longevity Sciences, Longitudinal Study of Aging (NILS-LSA). The NILS-LSA is a community-based random sample study of aging and age-related diseases that represents the total Japanese population of middle-aged and older adults. The lifestyle of residents of this area is typical of most individuals in Japan, and the participants are sex- and age-stratified. Participants aged 40–79 years at baseline (Wave 1: 1997–2000) were followed up every 2 years. Age and gender-matched random samples equivalent to the number of dropout participants were recruited, except for participants aged over 79 years, and male and female participants aged 40 years were also newly recruited every year. The examination intervals were as follows: Wave 2: 2000–2002, Wave 3: 2002–2004, Wave 4: 2004–2006, Wave 5: 2006–2008, Wave 6: 2008–2010, and Wave 7: 2010–2012. A reduced follow-up study was performed in Wave 8 from 2013 to 2016. Details of the NILS-LSA have been published elsewhere [18], and the protocol and basic data are provided on the relevant in web page (https://www.ncgg.go.jp/cgss/english/department/nils-lsa/). NILS-LSA is one of the largest cohort studies in Japan that assessed pure-tone audiometry.

From all the participants in the NILS-LSA, we included those who had moderate hearing loss in this study. The baseline of this study was the time when the participants first showed moderate hearing loss (details are explained below) in any wave. The exclusion criteria were (a) age under 60 years at baseline (N = 25), (b) a history of dementia at baseline (N = 12), and (c) absence of any essential information, such as the results of the cognitive assessment or the data for confounders at baseline (N = 56). We did not exclude participants who had otological diseases or adjust history of otological diseases as confounder because missing data was not small (N = 50). Study profile and the final sample are presented in Fig 1 and Table 1. Time 1 represented the baseline, and Time 2–8 referred to the subsequent waves. For example, if the participant had moderate hearing loss in Wave 2 and participated in Wave 4 and Wave 6, Wave 4 was Time 3, and Wave 6 was Time 5. Thus, a total of 1192 observations from 407 participants who with moderate hearing loss were used for the analysis. The mean follow-up period was 4.5 ± 3.9 years, and the average number of measurements was 2.9 ± 1.7. There was a total of 128 participants who used HA in this study. Of those, there were 66 participants who had moderate hearing loss and used HA since their first participation, however, the time at which they had moderate hearing loss and began using HA was unknown. The average time interval between baseline and report of HA use was 5.9 ± 3.1 years in the other 62 HA users.

### Evaluations

**Cognitive assessment.** Cognitive function was assessed using the Japanese version of the Wechsler Adult Intelligence Scale-Revised Short Forms (WAIS-R-SF) [19]. The WAIS-R-SF consists of four subtests; Information, Similarities, Picture completion, and Digit Symbol Substitution (DSST) [20]. The Information subtest assesses general knowledge by asking 29 general knowledge questions (possible scores range from 0 to 29). The Similarities subtest assesses logical abstract thinking by asking the participants to state the similarities among 14 items (possible scores range from 0 to 28). The Picture completion subtest assesses visual perception and memory by asking participants to point out the missing elements in a series of drawings (possible scores range from 0 to 21). The DSST assesses processing speed by asking participants to write as many symbols as possible that correspond to a given number in 90 s (possible

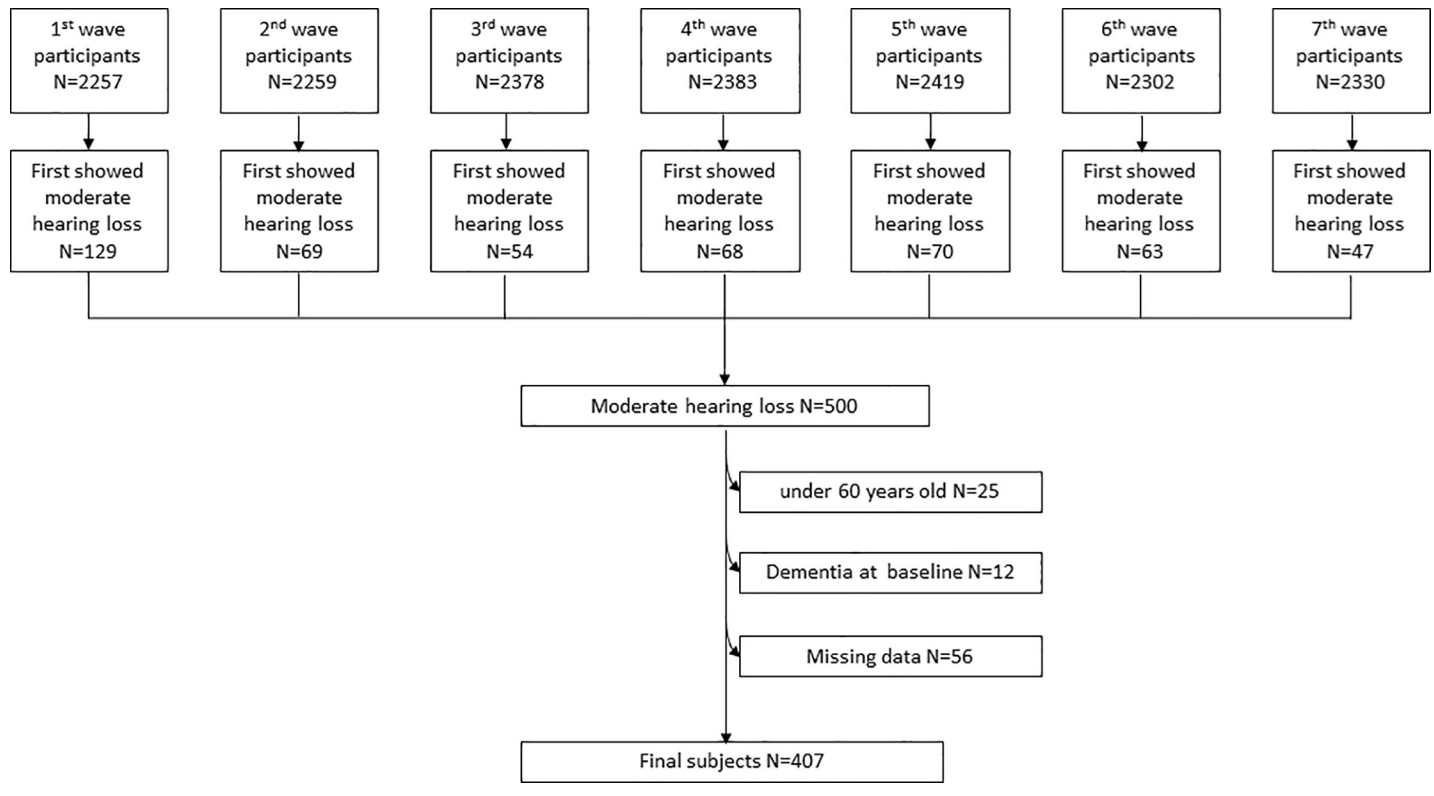

**Fig 1. Study profile.**

scores range from 0 to 93). Trained examiners (clinical psychologists or psychology graduate students) administered the test to each participant according to the standard instructions. The examiners spoke to the participants loudly, clearly, and slowly enough to understand. If the participants had HA, they wore them. Participants who could not understand the conversation adequately despite the above conditions were not included in the test.

**Hearing assessment and other measures.**   Air-conduction pure-tone audiometry was performed in a sound-proof booth by trained technicians using a diagnostic audiometer (AA-73A and AA-78; Rion, Tokyo, Japan). The pure-tone average threshold levels at four

**Table 1. Number of participants and durations of the follow-up periods from baseline.**

|  | N | Follow-up years from baseline | HA user, N | % |
|---|---|---|---|---|
| Baseline (Time 1) | 407 |  | 66 | 16.2 |
| Time 2 | 302 | 2.2 ± 0.4 | 70 | 23.2 |
| Time 3 | 200 | 4.2 ± 0.5 | 50 | 25.3 |
| Time 4 | 126 | 6.4 ± 0.5 | 45 | 35.7 |
| Time 5 | 80 | 8.4 ± 0.6 | 32 | 40.0 |
| Time 6 | 47 | 10.5 ± 0.7 | 26 | 55.3 |
| Time 7 | 23 | 12.5 ± 0.8 | 14 | 60.9 |
| Time 8 | 7 | 15.5 ± 0.7 | 3 | 42.9 |

The study population consisted of 407 participants with moderate hearing loss (pure-tone average threshold level of the better hearing ear at frequencies of 500, 1000, 2000, and 4000 Hz: PTABHE $\geq$ 40 and $<$ 70 dB).

frequencies (500, 1000, 2000, and 4000 Hz) with the better hearing ear (PTABHE) were used to determine hearing status. PTABHE $\geq$ 40 and $<$ 70 dB was defined as moderate hearing loss according to the definition of the Japan Audiological Society. We defined hearing status according to the WHO hearing impairment grade [21] in our previous studies [6, 17]; however, the WHO recently proposed a new grading system categorizing hearing impairment as follows; normal ($<$ 20 dB), mild (20–34 dB), moderate (35–49 dB), moderately severe (50–64 dB), severe (65–79 dB), and profound ($\geq$ 80 dB) based on the average threshold at 500, 1000, 2000, and 4000 Hz [22]. In Japan, the quadrant method ((500 Hz + 1000 Hz + 1000 Hz + 2000 Hz)/4) has been used for the diagnosis of hearing impairment for a long time, and if the hearing in both ears is 70 dB or more, or if the hearing in one ear is 90 dB or more and that in the other ear is 50 dB or more, the patient is diagnosed with an authorized hearing disability and can receive public assistance such as hearing aid subsidy. Under this background, the Japan Audiological Society defined moderate hearing impairment as a 40–69 dB average threshold of 500, 1000, 2000, and 4000 Hz. Thus, we conducted a study based on the diagnostic criteria of the Japan Audiological Society.

Participants answered a series of questionnaires, which included detailed assessments of their medical history and lifestyle. All participants were asked about their HA usage, and the possible answers were "always use", "sometimes use", "have a hearing aid, but never use it", and "have no hearing aid". The "always use" and "sometimes use" responses were considered to indicate HA usage. The responses for medical histories of dementia, hypertension, dyslipidemia, diabetes, ischemic heart disease, and stroke were categorized into "yes" or "no". Participants were also asked about their household income, occupation, marital status, smoking status, and education. Household income was divided into "greater than or equal to 5,500,000 yen/year" or "less than 5,500,000 yen/year." Occupation was divided into "having an occupation" or "unemployed." Marital status was divided into "married" or "unmarried." Smoking status was classified as "non-smoker" or "ex- or current smoker."

The participants' height and weight were measured, and obesity was defined as a body mass index greater than 25.0 kg/m$^2$. The Center for Epidemiologic Studies Depression Scale (CES-D) [23] was also assessed, and depressed mood was defined by a CES-D score over 15.

**Ethical approval.** The study protocol complied with the Declaration of Helsinki and was approved by the Committee on Ethics of Human Research of the National Institute for Longevity Sciences (No. 1369). Written informed consent was obtained from all participants.

**Statistical analysis.** Categorical variables are displayed as counts and percentages, and continuous variables are presented as mean ± standard deviation (SD) unless otherwise stated. A t-test was used to evaluate the differences in continuous variables, and the chi-square test was used to evaluate differences in categorical variables in univariate analysis of baseline data between groups with and without HA.

General linear mixed models were used to evaluate the associations between HA use and cognitive changes during the follow-up period. This model is similar to ordinary regression analysis, but because it allows correlation between the observations, it can handle missing data in repeated measurements more appropriately [24]. In this study, the model included fixed terms for the intercept which was baseline performance with a value of zero for all predictors, time (in years since baseline), HA use (no HA use at any time of follow-up period, or HA use), and the HA use × time interaction term. Covariates were age, sex, education, marital status, occupation, income, depressed mood, smoking status, obesity, and history of disease (hypertension, diabetes, dyslipidemia, ischemic heart disease and stroke) at baseline. The age at baseline was centered at the average age of the baseline for efficient convergence. We also calculated the random effects of intercept and slope using an unstructured covariance matrix, and predicted 12-year changes by considering potential confounders to make it easier to

compare with our previous study [6]. In addition, the analyses excluding Time 6–8 were performed because the follow-up rates were under 20% from Time 5.

Statistical analyses were conducted using the Statistical Analysis System (SAS) version 9.3 (SAS Institute, Cary, NC, USA). The two-sided significance level was set at p < 0.05.

## Results

The baseline characteristics of the participants are presented in Table 2. Among 407 older adults with moderate hearing loss, 279 (68.6%) did not use HA during the follow-up, and 128 (31.4%) used HA. The participants who used HA were significantly younger, had worse in PTABHE and higher education, and showed less depressed mood than non-users. The number of measurements, that is, the frequency of participation in the survey, was significantly higher among HA users. The baseline scores for Picture completion and DSST were significantly higher in HA users than in non-users, although there were no significant differences in Information and Similarities scores.

Table 3 shows the results of multivariable analyses to evaluate the changes in WAIS-R-SF scores during follow-up using the general linear mixed models. Model 1 was adjusted with minimally covariates including age, sex, and education year. Model 2 was additionally adjusted for PTABTE, history of disease (hypertension, diabetes, dyslipidemia, ischemic heart disease and stroke), smoking status, marital status, occupation, obesity, depression, and income. The main effect of time on Similarities was not significant in model 1 (p = 0.144), however, it

**Table 2. Participant characteristics at baseline.**

| | Total | No hearing aid usage | Hearing aid users | p-value |
|---|---|---|---|---|
| N | 407 | 279 | 128 | |
| Number of measurements | 2.9 ± 1.7 | 2.6 ± 1.5 | 3.7 ± 1.9 | < .0001 |
| Age, years | 74.6 ± 5.3 | 75.2 ± 5.2 | 73.4 ± 5.4 | 0.001 |
| PTABHE, dB | 45.0 ± 5.8 | 43.6 ± 4.5 | 48.2 ± 7.1 | < .0001 |
| Education, years | 10.5 ± 2.5 | 10.2 ± 2.5 | 11.1 ± 2.5 | 0.001 |
| Sex, male (n, %) | 279 (68.6%) | 175 (62.7%) | 90 (70.3%) | 0.136 |
| Marital status, married (n, %) | 327 (80.3%) | 220 (78.9%) | 107 (83.6%) | 0.264 |
| Occupation, employed (n, %) | 89 (21.9%) | 61 (21.9%) | 28 (21.9%) | 0.998 |
| Hypertension (n, %) | 186 (45.7%) | 123 (44.1%) | 63 (49.2%) | 0.335 |
| Dyslipidemia (n, %) | 86 (21.1%) | 55 (19.7%) | 31 (24.2%) | 0.301 |
| Diabetes (n, %) | 61 (15.0%) | 43 (15.4%) | 18 (14.1%) | 0.723 |
| Ischemic heart disease (n, %) | 51 (12.5%) | 37 (13.3%) | 14 (10.9%) | 0.511 |
| Stroke (n, %) | 35 (8.6%) | 27 (9.7%) | 8 (6.3%) | 0.252 |
| Smoker or ex-smoker (n, %) | 217 (53.3%) | 146 (52.3%) | 71 (55.5%) | 0.556 |
| Obesity (n, %) | 75 (18.4%) | 53 (19.0%) | 22 (17.2%) | 0.662 |
| High income (≥5,500,000 yen/year) (n, %) | 111 (27.3%) | 69 (24.7%) | 42 (32.8%) | 0.089 |
| Depressed mood (>15 CES-D score) (n, %) | 76 (18.7%) | 63 (22.6%) | 13 (10.2%) | 0.003 |
| Information (maximum score, 29) | 12.9 ± 5.5 | 12.8 ± 5.7 | 13.2 ± 5.3 | 0.542 |
| Similarities (maximum score, 28) | 10.1 ± 5.4 | 10.0 ± 5.4 | 10.5 ± 5.4 | 0.377 |
| Picture completion (maximum score, 21) | 9.5 ± 3.9 | 9.1 ± 4.0 | 10.3 ± 3.8 | 0.006 |
| Digit symbol substitution (maximum score, 93) | 38.1 ± 10.9 | 37.0 ± 10.7 | 40.5 ± 11.0 | 0.003 |

p-values were calculated using the chi-square test for categorical values and t-test for continuous values.

PTABTE: pure-tone average threshold level of the better ear at frequencies of 500, 1000, 2000, and 4000 Hz

CES-D: The Center for Epidemiologic Studies Depression Scale

**Table 3. Results of general linear mixed model for WAIS-R-SF scores.**

| | Explanatory variables | Information | | | Similarities | | | Picture completion | | | Digit symbol substitution (DSST) | | |
|---|---|---|---|---|---|---|---|---|---|---|---|---|---|
| | | Estimate | SE | p-value | Estimate | SE | p-value | Estimate | SE | p-value | Estimate | SE | p-value |
| Model 1 | Time | -0.164 | 0.034 | < .0001 | -0.064 | 0.044 | 0.144 | 0.059 | 0.034 | 0.079 | -0.591 | 0.075 | < .0001 |
| | Usage of hearing aid | -0.444 | 0.514 | 0.388 | -0.185 | 0.490 | 0.706 | 0.637 | 0.379 | 0.093 | 0.714 | 1.032 | 0.490 |
| | Time × usage of hearing aid | 0.104 | 0.048 | 0.033 | -0.019 | 0.062 | 0.755 | -0.035 | 0.047 | 0.457 | -0.156 | 0.110 | 0.160 |
| Model 2 | Time | -0.163 | 0.034 | < .0001 | -0.061 | 0.044 | 0.013 | 0.059 | 0.034 | 0.101 | -0.586 | 0.075 | < .0001 |
| | Usage of hearing aid | -0.066 | 0.546 | 0.904 | 0.471 | 0.521 | 0.366 | 1.616 | 0.394 | < .0001 | 2.214 | 1.091 | 0.043 |
| | Time × usage of hearing aid | 0.100 | 0.048 | 0.040 | -0.031 | 0.062 | 0.614 | -0.040 | 0.047 | 0.404 | -0.169 | 0.109 | 0.122 |

Model 1 was adjusted for age, sex, education year.

Model 2 was adjusted for the covariates in model 1 + PTABTE, history of hypertension, history of diabetes, history of dyslipidemia, history of stroke, history of ischemic heart disease, smoking status, marital status, occupation, obesity, depression, and income.

PTABTE: pure-tone average threshold level of the better ear at frequencies of 500, 1000, 2000 and 4000 Hz

became significant after adjusted various covariates (p = 0.013). In final model, the main effect of time on Picture completion was not significant (p = 0.101 in model 2), although the effects on other scores were significant (Information and DSST, p < .0001 in model 2). The effects of usage of HA on Picture completion and DSST became significant after adjusting for age, sex, education, marital status, occupation, income, depressed mood, smoking status, obesity, and history of disease (hypertension, dyslipidemia, diabetes, ischemic heart disease, and stroke). There was no difference in the Information score at baseline in the HA use group and no HA use group, however, the HA use × time (follow-up years) interaction was significant after adjusting for confounders, and the p-values were 0.033 in model 1 and 0.040 in model 2.

The predicted models of the 12-year change in WAIS-R-SF scores with or without HA use are shown in Fig 2. The Information score in HA users declined slower than non-users, although there was no difference in baseline scores. The scores of Similarities declined significantly only in HA users, although there was no significant difference in the rate of change between HA users and non-users. There were no significant changes in Picture completion scores over time in either the HA users or non-users. The DSST scores declined significantly in both HA users and non-users, and there was no significant difference in the rate of change between HA users and non-users.

The Results of general linear mixed analyses excluded the data of Time 6–8 are shown in S1 Table. The HA x time interaction of the Information score was significant when the analysis was performed excluding Time 8, however, it was no longer significant after excluding Time 8 and 7. The main effect of HA use on the scores of Picture completion and DSST tended to be kept.

## Discussion

In this longitudinal study, we found that participants with moderate hearing loss who used HA retained their Information scores, whereas the Information score declined in those who did not use HA. We had previously investigated the effect of hearing loss over 25 dB on the WAIS-R-SF scores in participants aged 60–79 years at baseline [6]. The hearing loss × time interaction was significant for the Information and DSST scores. Studies investigating the effect of age on WAIS scores have demonstrated that Information and Similarities scores have stability during aging [25, 26], although the tolerance of the Information score was lost in hearing-impaired individuals in a previous study [6]. In the present analysis, this decline in the

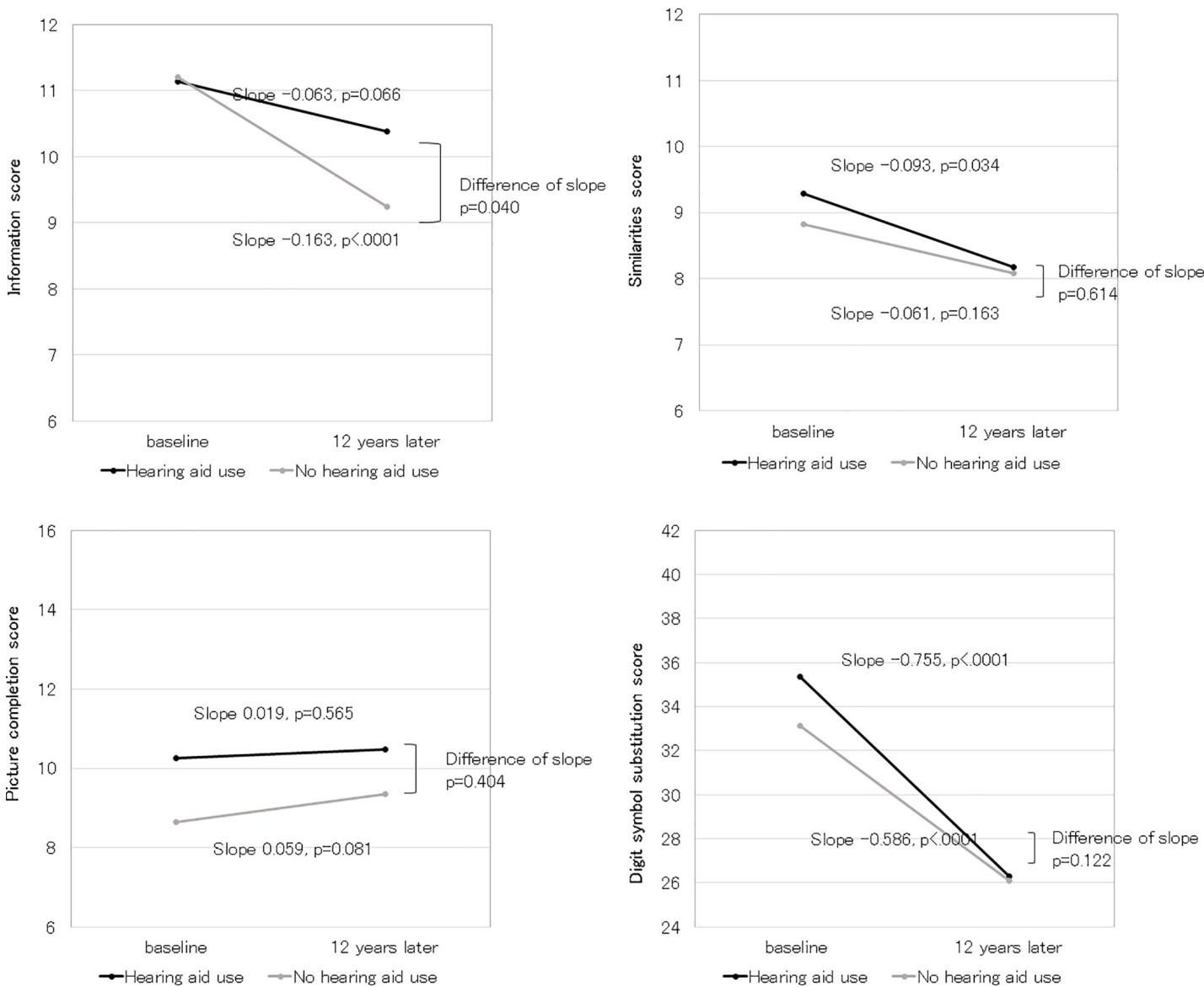

**Fig 2. Model-predicted 12-year change in cognitive ability by hearing aid use in participants with moderate hearing loss.** A. Result of Information. B. Result of Similarities. C. Result of Picture completion. D. Result of Digit symbol substitution. General linear mixed model adjusted for age, sex, PTABHE, education year, history of hypertension, history of dyslipidemia, history of diabetes, history of ischemic heart disease, history of stroke, smoking status, obesity, marital status, income, depression, and occupation at baseline. PTABTE: pure-tone average threshold level of the better ear at frequencies of 500, 1000, 2000, and 4000 Hz.

Information score was suppressed in the HA using group. The retention of the Information score may be attributed to the fact that HA use allowed participants to continue acquiring verbal information. In other words, people with moderate hearing loss may be more vulnerable to inadequate information. In contrast, we observed that the decline in the DSST score was significantly steeper in the hearing-impaired group than in hearing participants in our previous study [6], although HA use showed no significant longitudinal association with DSST score changes in this study. The DSST measures processing speed and is considered to be highly reflective of aging. Therefore, DSST was sensitive in revealing the correlation of hearing loss and cognitive function in some cross-sectional studies [27, 28], and Lin reported that HA use was significantly associated with higher DSST scores. However, in longitudinal studies, the

association between hearing loss and the DSST score has been controversial [29, 30]. Further-more, some reports have stated that HA use had no significant effect on changes in the DSST score [31, 32], whereas a recent study reported that HA use improved the DSST score [33]. The DSST is sensitive to the presence of cognitive dysfunction and changes in cognitive function across a wide range of clinical populations, including dementia, depression, schizophre-nia, and sleep disorder [34]. The fact that DSST is sensitive to such conditions may explain these controversial results. In contrast to the DSST, scores of Picture completion did not decrease after covariates were adjusted, furthermore, they tended to increase in non-HA users. It has been suggested that the Picture completion might be resilient to the impact of brain damage [35], and be maintained better with age than most other performance tests [20]. Thus, we thought scores of Picture completion did not decrease after adjusting for covariates such as practice effects [36].

We also found that participants with moderate hearing loss who used HA had higher base-line Picture completion and DSST scores than those who did not use HA, although the changes in these scores over time did not significantly differ from those in non-HA users. It is reported that the Information and Similarities subtests are verbal tests reflecting crystalized intelligence that is accumulated through life experiences and education, while Picture completion and DSST subtests are performance tests reflecting fluid intelligence that is sensitive to aging [25]. Fluid intelligence is the ability to solve problems when adapting to new learning and is an important resource for dealing effectively with new experiences [37]. The results of the tests reflecting fluid intelligence were high at baseline, indicating that participants with high fluid intelligence tended to use HA actively.

Some review articles have pointed out three dominant hypotheses underlying the relation-ship between hearing and cognition: (a) the common cause hypothesis, (b) the cognitive load hypothesis, and (c) the cascade hypothesis [38–40]. The common cause hypothesis suggests that common factors, such as genetic factors, microvascular insufficiency, or oxidative stress cause both hearing loss and cognitive decline. In this model, hearing loss and cognitive func-tion change in parallel, and the causality is unclear. The cognitive load hypothesis postulates that degraded auditory input due to hearing loss places an increased demand for limited pro-cessing resources. The cascade hypothesis can be further divided into two hypotheses: one the-ory states that auditory deprivation causes reallocation of cognitive resources, which leads to cognitive decline, while the second theory suggests that problems such as inactivity, depres-sion, and social isolation caused by hearing loss accelerate cognitive decline. These hypotheses may be intricately involved. The different influences of hearing and HA use on each cognitive domain, such as the results of the four tests of WAIS-R-SF, may reflect this complexity.

The HA usage rate among participants with moderate hearing loss was 31.4% in Japanese community-dwelling older adults aged 60 years or more. The HA using participants were sig-nificantly younger and had worse PTABTE, higher education levels, and less depressed mood than the participants who did not use HA. Worse PTABHE and high education have been reported to be important factors for HA use [41] or HA acquisition [42] in the Epidemiology of Hearing Loss Study. Participants with longer education periods had a higher income; more-over, they would manage their health and use the devices more effectively. They may also be more motivated to wear HA. The frequency of participation in the survey was significantly higher among HA users. Hearing loss has been reported to increase the likelihood of clinically relevant depression symptoms in the older adult population, although the effect of HA use on depressed symptoms is controversial [43]. In this study, it was unclear whether participants did not use HA because they had depression, or whether participants who used HA were less likely to get depressed. However, it is presumed that many HA users were motivated because they frequently participated in the study. We previously reported that openness has a

protective effect against the decline in Information and Similarities scores [44]. In this analysis, we could not adjust for openness because openness was assessed only in the 2nd and 7th waves. Thus, it is possible that the background of HA users influences this result more than the use of HA itself, although we adjusted for the confounders such as age, sex, education, marital status, occupation, income, depressed mood, smoking status, obesity, and history of disease.

This study had some limitations. Since we set the baseline at the point where moderate hearing loss was detected, the timing of starting HA use was different for each participant. Furthermore, we could not evaluate the HA fitting situation. HA has long been available to the consumer through internet or retail outlets in Japan, without the need of a hearing test to purchase. It was reported that 17.4% of HA owners purchased them through internet or mail order [14], and personal sound amplification products (PSAPs) and HA are sometimes not distinguished. Thus, some participants may treat PSAPs as HA. Second, we analyzed participants aged 60 years or more owing to the small number of participants with moderate hearing loss under the age of 60 years. However, the Lancet Commission on Dementia Prevention, Intervention and Care noted that midlife hearing loss is one of modifying risk factors [4]. Thus, it is necessary to continue to study the effects of HA on middle-aged people with hearing loss. Third, our study sample size was relatively small, and we could not adjust the data with all of the relevant covariates due to missing data. In addition, the significant difference of the Information score's change was not observed when the follow-up period was limited. However, NILS-LSA is one of the largest studies that evaluates pure-tone audiometry in Japan, and the difference in the impact of HA on each subtest is important for considering the association between hearing and cognition.

## Conclusions

We found that participants with moderate hearing loss who used HA for an average period of 4.5 years retained their Information score, whereas the score declined in those who did not use HA in this longitudinal study. The 12-year slopes did not differ between HA users and non-users for the Similarities, Picture Completion and DSST scores. Our study showed that HA use could have a protective effect against the decline in general knowledge in older adults with moderate hearing loss.

## Supporting information

**S1 Table. Results of general linear mixed model for WAIS-R-SF scores excluding data from Times 6–8; 7–8 and 8.** Adjusted for age, sex, PTABHE, education year, history of hypertension, history of dyslipidemia, history of diabetes, history of ischemic heart disease, history of stroke, smoking status, obesity, marital status, income, depression, and occupation at baseline.
(DOCX)

## Acknowledgments

We thank all of the participants and our colleagues in the NILS-LSA.

## Author Contributions

**Conceptualization:** Saiko Sugiura, Yukiko Nishita.

**Data curation:** Yukiko Nishita, Chikako Tange, Rei Otsuka, Fujiko Ando, Hiroshi Shimokata.

**Formal analysis:** Saiko Sugiura, Yukiko Nishita.

**Funding acquisition:** Hirokazu Suzuki, Rei Otsuka.

**Investigation:** Saiko Sugiura, Yukiko Nishita, Chikako Tange, Rei Otsuka, Fujiko Ando, Hiroshi Shimokata.

**Project administration:** Rei Otsuka, Fujiko Ando, Hiroshi Shimokata.

**Supervision:** Yasue Uchida, Masaaki Teranishi, Tsutomu Nakashima, Rei Otsuka, Fujiko Ando, Hiroshi Shimokata.

**Writing – original draft:** Saiko Sugiura.

**Writing – review & editing:** Yukiko Nishita, Yasue Uchida, Mariko Shimono, Hirokazu Suzuki, Masaaki Teranishi, Tsutomu Nakashima, Chikako Tange, Rei Otsuka, Fujiko Ando, Hiroshi Shimokata.

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
