## [Decision Letter · Decision Letter 0]

11 May 2021

PONE-D-21-13516

Longitudinal effects of hearing aid usage on cognition in community-dwelling Japanese older adults with moderate hearing loss.

PLOS ONE

Dear Dr. Sugiura,

Thank you for submitting your manuscript to PLOS ONE. After careful consideration, we feel that it has merit but does not fully meet PLOS ONE’s publication criteria as it currently stands. Therefore, we invite you to submit a revised version of the manuscript that addresses the points raised during the review process.

The manuscript does not reach to an enough level for the acceptance in the Journal. 

See the Reviewers' comments carefully and respond them appropriately.

We look forward to receiving your revised manuscript.

Kind regards,

Masaki Mogi

Academic Editor

PLOS ONE

Journal Requirements:

Reviewers' comments:

Reviewer's Responses to Questions

**Comments to the Author**

1. Is the manuscript technically sound, and do the data support the conclusions?

Reviewer #1: Yes

Reviewer #2: Yes

2. Has the statistical analysis been performed appropriately and rigorously? 

Reviewer #1: Yes

Reviewer #2: Yes

3. Have the authors made all data underlying the findings in their manuscript fully available?

Reviewer #1: Yes

Reviewer #2: No

4. Is the manuscript presented in an intelligible fashion and written in standard English?

Reviewer #1: No

Reviewer #2: Yes

5. Review Comments to the Author

Reviewer #1: This epidemiologic study focused on the protective effect of hearing aids in older adults. The experiment itself is carried out with careful treatments, but the reviewer thinks the sample size is too small.

Methods

Why were the participants under 60 years old excluded?

Please explain the reason. Usually, the definition of old age is over 65 years old.

Table 1

This table was not informative. Please create a flow-chart.

The number of participants on each group should be shown.

Table 1

The follow-up rates were under 20% from “Time 6”. How were the results changed, if the analysis was performed until “Time5”?

Methods

Were the participants with otological diseases, like otitis media excluded? Please explain it.

Methods

The reviewer thought that the covariates were not enough in this study. The previous report which the authors published on Auris Nasus Larynx included more covariates than this study. Please include more covariates and re-calculate the data. Obesity, history of hyperlipidemia, alcohol intake and history of cardiovascular disease, like stroke and ischemic heart disease are essential covariates, when the cognitive functions are analyzed.

Results

Why were the picture completion scores higher than baseline on both groups? Are the picture completion tests not useful tests on evaluating cognitive function? Please explain it.

Results

Only one model was shown on Table 3. Please create more than two models.

Discussion from 297 to 308

This paragraph should be stated at method section.

Abbreviations about hearing aids were incorrect like “Has” at some points. Please correct them to “HA”.

Reviewer #2: This is a well-conducted study of the association of hearing aid usage with longitudinal cognitive function among 407 older adults with moderate hearing impairment. Given the prevalence of hearing impairment, and prevalence of cognitive decline with age, this is an important and relatively under-studied area. This study represents an important contribution to the literature. Strengths include the relatively large number of participants; objective measures of hearing thresholds; control for many important potentially confounding factors; longitudinal measures, and use of general linear mixed effects regression models. Limitations are adequately noted. However, I have several suggestions for improvement:

Title: Since this is an observational study that describes the association between hearing aid usage and cognitive decline it is important to avoid language the implies causation. Thus the word “Effects” should be changed to “Associations” in the title and throughout the text.

For the full implication of the results to be appreciated from the abstract, it is important to state that slopes did not differ between HA users and non-users for the Similarities, Picture Completion and Digit Symbol Substitution tests.

Line 49 include “is”: The global population of older adults is expected to reach 1.4 billion by 2030 and 2.1 billion by 2050.

Line 88: avoid causal language such as “HAS use influenced changes…” since differences between HA users and non-users may arise from uncontrolled confounding.

In a few places in the manuscript, hearing aid use is sometimes abbreviated as “ha” or “has” – rather than HA or HAs.

It would be helpful if Table 1 was stratified by hearing aid use and non-hearing aid use to allow the reader to appreciate the difference in participation/follow-up time between these two groups.

It would be helpful to report on the average time interval between baseline and report of HA use, as well as the range of that interval. This is important to appreciate how much time may have passed between the onset of hearing impairment and beginning of use of hearing aids, which could affect associations with cognitive function.

Line 312-313. It is unclear why being a local resident would affect whether participants considered PSAPs as HAs. It is not clear whether the authors are implying that participants may have indicated HA use when they were not using HA, but were using PSAPs? "In addition, personal sound amplification products (PSAPs) may be treated as HAs, because the participants were local residents so they may not be able to distinguish HA from PSAPs”.

Conclusion: As with the abstract, it is important to note the implications of the non-significant findings as well. That is, these results shows that HA usage is not associated with reduction in rate of cognitive decline in some tasks that show significantly higher rates of decline among individuals with hearing impairment relative to those without hearing impairment.

*Authors indicate that they are unable to share their data due to third party restrictions.

6. PLOS authors have the option to publish the peer review history of their article (what does this mean?). If published, this will include your full peer review and any attached files.

Reviewer #1: No

Reviewer #2: No

---

## [Author Response · Author response to Decision Letter 0]

3 Sep 2021

Our incorporation of the Reviewer comments is as follows:   

  

The changed part in the manuscript was shown in red color.

To reviewer #1: This epidemiologic study focused on the protective effect of hearing aids in older adults. The experiment itself is carried out with careful treatments, but the reviewer thinks the sample size is too small.

→Thank you for reviewing our manuscript. As you pointed out, the sample size was not large. However, NILS-LSA is one of the largest cohort studies in Japan that assessed pure-tone audiometry, and we tried to increase the sample size by collecting and reconstructing the samples with moderate hearing loss from all participants. We mentioned this limitation in discussion. (Line 107-110, 345-350)

Methods

Why were the participants under 60 years old excluded?

Please explain the reason. Usually, the definition of old age is over 65 years old.

→In Japan, people over 60 have been regarded as elderly and retired. Of course, the current situation is that they are being reviewed to make them over 65. Also, the NILS-LSA was designed to have the same number of participants in each decade (40s, 50s, 60s, 70s). Thus, we did not change the definition of old age as over 65 years old in accordance with our previous study.

Table 1

This table was not informative. Please create a flow-chart.

The number of participants on each group should be shown.

→We added a flow-chart (Fig 1). We also added the number of HA users to Table 1 as the reviewer#2 suggested.

Table 1

The follow-up rates were under 20% from “Time 6”. How were the results changed, if the analysis was performed until “Time5”?

→We performed additional analyses excluding Time 6-8, and added these results as S1 Table. (Line 200, 201, 261-264)

Methods

Were the participants with otological diseases, like otitis media excluded? Please explain it.

→We did not exclude the participants who had the history of ontological diseases because there were 50 participants of 407 whose data was missing. We added the explanation and discussed it. (Line 114, 115, 345-350) 

Methods

The reviewer thought that the covariates were not enough in this study. The previous report which the authors published on Auris Nasus Larynx included more covariates than this study. Please include more covariates and re-calculate the data. Obesity, history of hyperlipidemia, alcohol intake and history of cardiovascular disease, like stroke and ischemic heart disease are essential covariates, when the cognitive functions are analyzed.

→Choice of covariates is difficult problem. We added obesity, history of hyperlipidemia, and history of ischemic heart disease as suggested. We did not add alcohol intake because there were 48 participants of 407 who lack an answer for the questionnaire about alcohol intake. Again, we discussed this limitation. (Line 345-350) 

Results

Why were the picture completion scores higher than baseline on both groups? Are the picture completion tests not useful tests on evaluating cognitive function? Please explain it.

→It has been suggested that the Picture completion might be resilient to the impact of brain damage (Kaufman AS & Lichtenberger EO, Assessing adolescent and adult intelligence. 3rd ed. Hoboken, NJ: John Wiley & Sons, 2006), and to hold up with age better than most other performance tests (Wechsler D. The measurement of adult intelligence. 3rd et. Baltimore, OH: The Williams & Wilkins Company, 1944). Thus, we thought scores of Picture completion did not decrease after adjusting for covariates such as practice effects. We discussed it in previous study (Niahita Y, et al. Does high educational level protect against intellectual decline in older adults? : A 10-year longitudinal study. Jpn Psychol Res 2013;55:378-389). We added the reference and mentioned it. (Line 246, 247, 289-294)

Results

Only one model was shown on Table 3. Please create more than two models.

→We create two models, model 1 is adjusted with covariates same as our previous study (Uchida Y, et al. The longitudinal impact of hearing impairment on cognition differs according to cognitive domain. Front Aging Neurosci 2016;8:201), and model 2 is adjusted with covariates model 1 + depression, income, obesity, and history of dyslipidemia.

Discussion from 297 to 308

This paragraph should be stated at method section.

→We moved this paragraph to method section. (Line 154-165)

Abbreviations about hearing aids were incorrect like “Has” at some points. Please correct them to “HA”.

→We corrected as suggested.

To reviewer #2: This is a well-conducted study of the association of hearing aid usage with longitudinal cognitive function among 407 older adults with moderate hearing impairment. Given the prevalence of hearing impairment, and prevalence of cognitive decline with age, this is an important and relatively under-studied area. This study represents an important contribution to the literature. Strengths include the relatively large number of participants; objective measures of hearing thresholds; control for many important potentially confounding factors; longitudinal measures, and use of general linear mixed effects regression models. Limitations are adequately noted. However, I have several suggestions for improvement:

→We deeply appreciate your kind comments.

Title: Since this is an observational study that describes the association between hearing aid usage and cognitive decline it is important to avoid language the implies causation. Thus the word “Effects” should be changed to “Associations” in the title and throughout the text.

→We changed as suggested.

For the full implication of the results to be appreciated from the abstract, it is important to state that slopes did not differ between HA users and non-users for the Similarities, Picture Completion and Digit Symbol Substitution tests.

→We added a sentence as suggested. (Line 44, 45)

Line 49 include “is”: The global population of older adults is expected to reach 1.4 billion by 2030 and 2.1 billion by 2050.

→We changed as suggested. (Line 51)

Line 88: avoid causal language such as “HAs use influenced changes…” since differences between HA users and non-users may arise from uncontrolled confounding.

→We changed as suggested. (Line 89, 90)

In a few places in the manuscript, hearing aid use is sometimes abbreviated as “ha” or “has” – rather than HA or HAs.

→We corrected as suggested.

It would be helpful if Table 1 was stratified by hearing aid use and non-hearing aid use to allow the reader to appreciate the difference in participation/follow-up time between these two groups.

→We added the number of HA users to Table 1 as suggested.

It would be helpful to report on the average time interval between baseline and report of HA use, as well as the range of that interval. This is important to appreciate how much time may have passed between the onset of hearing impairment and beginning of use of hearing aids, which could affect associations with cognitive function.

→There were 66 participants who had moderate hearing loss and use HA from the first participation, and we could not know when they began to use HA. So, we added the average time interval between baseline and report of HA use in remaining 62 participants. (Line 121-124)

Line 312-313. It is unclear why being a local resident would affect whether participants considered PSAPs as HAs. It is not clear whether the authors are implying that participants may have indicated HA use when they were not using HA, but were using PSAPs? "In addition, personal sound amplification products (PSAPs) may be treated as HAs, because the participants were local residents so they may not be able to distinguish HA from PSAPs”.

→Opposite to the United States, HA has been available to the consumer through Internet or retail outlets, for example, eyeglass shop, in Japan for a long time. Also, pre-purchase hearing evaluation is not necessary to buy HA. So, people sometimes mistake PSAPs as HA in Japan. We added a sentence explaining the above. (Line 337-341)

Conclusion: As with the abstract, it is important to note the implications of the non-significant findings as well. That is, these results shows that HA usage is not associated with reduction in rate of cognitive decline in some tasks that show significantly higher rates of decline among individuals with hearing impairment relative to those without hearing impairment.

→We added a sentence as suggested. (Line 355, 356)

We corrected some errors, and added one recent study (Line 285, 286, Reference number 33).

---

## [Decision Letter · Decision Letter 1]

21 Sep 2021

PONE-D-21-13516R1Longitudinal associations between hearing aid usage and cognition in community-dwelling Japanese older adults with moderate hearing loss.PLOS ONE

Dear Dr. Sugiura,

Thank you for submitting your manuscript to PLOS ONE. After careful consideration, we feel that it has merit but does not fully meet PLOS ONE’s publication criteria as it currently stands. Therefore, we invite you to submit a revised version of the manuscript that addresses the points raised during the review process.

Minor revisions are still necessary in the present study. See the Reviewer's comments and respond them appropriately.

We look forward to receiving your revised manuscript.

Kind regards,

Masaki Mogi

Academic Editor

PLOS ONE

Journal Requirements:

Additional Editor Comments (if provided):

Reviewers' comments:

Reviewer's Responses to Questions

**Comments to the Author**

1. If the authors have adequately addressed your comments raised in a previous round of review and you feel that this manuscript is now acceptable for publication, you may indicate that here to bypass the “Comments to the Author” section, enter your conflict of interest statement in the “Confidential to Editor” section, and submit your "Accept" recommendation.

Reviewer #1: All comments have been addressed

Reviewer #2: (No Response)

2. Is the manuscript technically sound, and do the data support the conclusions?

Reviewer #1: Yes

Reviewer #2: Yes

3. Has the statistical analysis been performed appropriately and rigorously? 

Reviewer #1: Yes

Reviewer #2: Yes

4. Have the authors made all data underlying the findings in their manuscript fully available?

Reviewer #1: Yes

Reviewer #2: Yes

5. Is the manuscript presented in an intelligible fashion and written in standard English?

Reviewer #1: Yes

Reviewer #2: No

6. Review Comments to the Author

Reviewer #1: The reviewer thinks authors properly responded to the reviewers' comments and precisely revised the manuscript.

Reviewer #2: The authors have been responsive to prior reviewers and have improved the manuscript. I note below a few minor suggestions to improve clarity:

Abstract line 40 : remove “therefore”

Abstract lines 41 – 43: The following sentence is unclear: “The model-predicted 12-year slope

with centralizing age indicated that the Information score showed no significant decline in the participants using HA, although this tolerance was not found in the participants not using HA.” Do the authors mean that non-HA users showed significant decline in Information scores over the 12 year follow-up period whereas HA users did not? Since the slope in the HA group approached significance, I believe a better interpretation would be that the non HA users showed greater decline over time on Information scores than did HA users.

Results, pg 17, line 242 change were to are: “are shown in Fig 2”

Results, pg 18, line 261 change were to are: “are shown in S1 Table. Also, please change the table title to be more informative. Eg. “Results of general linear mixed model for WAIS-R-SF scores excluding data from Times 6-8; 7-8 and 8”

Results Line 246 – 247. This sentence is unclear: “The Picture completion scores did not decrease especially in the non-users, however, there was no significant difference in the rate of change between two groups”. This would be better expressed as “There were no significant changes in Picture completion scores over time in either the HA or non HA users.”

Table 3: I am not sure it is necessary to show the results from model 1 and model 2. There was no a priori reason for assessing differences between these models, although I understand that additional covariates were added in response to review. However, it would be more meaningful to show results of an unadjusted (or minimally adjusted model – including eg. age, sex, education) and the model with all the covariates (fully adjusted model).

7. PLOS authors have the option to publish the peer review history of their article (what does this mean?). If published, this will include your full peer review and any attached files.

Reviewer #1: No

Reviewer #2: No

---

## [Author Response · Author response to Decision Letter 1]

26 Sep 2021

Our incorporation of the Reviewer comments is as follows:     

The changed part in the manuscript was shown in red color.

To Reviewer #1: 

The reviewer thinks authors properly responded to the reviewers' comments and precisely revised the manuscript.

→Thanks to your kind review, our manuscript is better.

To Reviewer #2: 

The authors have been responsive to prior reviewers and have improved the manuscript. I note below a few minor suggestions to improve clarity:

→Thank you for reviewing our manuscript.

Abstract line 40 : remove “therefore”

→We corrected as suggested.

Abstract lines 41 – 43: The following sentence is unclear: “The model-predicted 12-year slope

with centralizing age indicated that the Information score showed no significant decline in the participants using HA, although this tolerance was not found in the participants not using HA.” Do the authors mean that non-HA users showed significant decline in Information scores over the 12 year follow-up period whereas HA users did not? Since the slope in the HA group approached significance, I believe a better interpretation would be that the non HA users showed greater decline over time on Information scores than did HA users.

→We corrected as suggested.

Results, pg 17, line 242 change were to are: “are shown in Fig 2”

→We corrected the sentence as suggested.

Results, pg 18, line 261 change were to are: “are shown in S1 Table. Also, please change the table title to be more informative. Eg. “Results of general linear mixed model for WAIS-R-SF scores excluding data from Times 6-8; 7-8 and 8”

→We corrected the sentence and changed the title as suggested.

Results Line 246 – 247. This sentence is unclear: “The Picture completion scores did not decrease especially in the non-users, however, there was no significant difference in the rate of change between two groups”. This would be better expressed as “There were no significant changes in Picture completion scores over time in either the HA or non HA users.”

→We corrected the sentence as suggested.

Table 3: I am not sure it is necessary to show the results from model 1 and model 2. There was no a priori reason for assessing differences between these models, although I understand that additional covariates were added in response to review. However, it would be more meaningful to show results of an unadjusted (or minimally adjusted model – including eg. age, sex, education) and the model with all the covariates (fully adjusted model).

→We changed the model 1 as minimally adjusted model. The sentences of results (Line 221-231) were also changed.

---

## [Decision Letter · Decision Letter 2]

30 Sep 2021

Longitudinal associations between hearing aid usage and cognition in community-dwelling Japanese older adults with moderate hearing loss.

PONE-D-21-13516R2

Dear Dr. Sugiura,

We’re pleased to inform you that your manuscript has been judged scientifically suitable for publication and will be formally accepted for publication once it meets all outstanding technical requirements.

Kind regards,

Masaki Mogi

Academic Editor

PLOS ONE

Additional Editor Comments (optional):

No further comment.

Reviewers' comments:

Reviewer's Responses to Questions

**Comments to the Author**

1. If the authors have adequately addressed your comments raised in a previous round of review and you feel that this manuscript is now acceptable for publication, you may indicate that here to bypass the “Comments to the Author” section, enter your conflict of interest statement in the “Confidential to Editor” section, and submit your "Accept" recommendation.

Reviewer #2: All comments have been addressed

2. Is the manuscript technically sound, and do the data support the conclusions?

Reviewer #2: (No Response)

3. Has the statistical analysis been performed appropriately and rigorously? 

Reviewer #2: (No Response)

4. Have the authors made all data underlying the findings in their manuscript fully available?

Reviewer #2: (No Response)

5. Is the manuscript presented in an intelligible fashion and written in standard English?

Reviewer #2: (No Response)

6. Review Comments to the Author

Reviewer #2: (No Response)

7. PLOS authors have the option to publish the peer review history of their article (what does this mean?). If published, this will include your full peer review and any attached files.

Reviewer #2: No

---

## [Editor Report · Acceptance letter]

4 Oct 2021

PONE-D-21-13516R2 

 Longitudinal associations between hearing aid usage and cognition in community-dwelling Japanese older adults with moderate hearing loss. 

Dear Dr. Sugiura:

I'm pleased to inform you that your manuscript has been deemed suitable for publication in PLOS ONE. Congratulations! Your manuscript is now with our production department. 

Kind regards, 

on behalf of

Dr. Masaki Mogi 

Academic Editor

PLOS ONE